# Clinicopathologic Features, Genetics, Treatment, and Long-Term Outcomes in Japanese Children and Young Adults with Benign Recurrent Intrahepatic Cholestasis: A Multicenter Study

**DOI:** 10.3390/jcm12185979

**Published:** 2023-09-15

**Authors:** Ken Kato, Shuichiro Umetsu, Takao Togawa, Koichi Ito, Takayoshi Kawabata, Teruko Arinaga-Hino, Naoya Tsumura, Ryosuke Yasuda, Yutaro Mihara, Hironori Kusano, Shogo Ito, Kazuo Imagawa, Hisamitsu Hayashi, Ayano Inui, Yushiro Yamashita, Tatsuki Mizuochi

**Affiliations:** 1Department of Pediatrics and Child Health, Kurume University School of Medicine, Kurume 830-0011, Japan; 2Department of Pediatric Hepatology and Gastroenterology, Saiseikai Yokohamashi Tobu Hospital, Yokohama 230-0012, Japan; 3Department of Pediatrics and Neonatology, Nagoya City University Graduate School of Medical Sciences, Nagoya 467-8601, Japan; 4Department of Pediatrics, Ishikawa Prefectural Central Hospital, Kanazawa 920-8530, Japan; 5Division of Gastroenterology, Department of Medicine, Kurume University School of Medicine, Kurume 830-0011, Japan; 6Department of Pathology, Kurume University School of Medicine, Kurume 830-0011, Japan; 7Department of Child Health, Institute of Medicine, University of Tsukuba, Tsukuba 305-8546, Japan; 8Laboratory of Molecular Pharmacokinetics, Graduate School of Pharmaceutical Science, The University of Tokyo, Tokyo 113-0033, Japan

**Keywords:** cholestasis, children, *ATP8B1*, *ABCB11*, rifampicin, cholestyramine

## Abstract

Background: Few reports of benign recurrent intrahepatic cholestasis (BRIC) have focused on East Asian patients. We describe the clinicopathologic features, genetics, treatment, and outcomes in Japanese BRIC patients. Methods: We recruited patients with BRIC type 1 (BRIC-1) or 2 (BRIC-2) treated at four pediatric centers and one adult center between April 2007 and March 2022. Demographics, clinical course, laboratory results, molecular genetic findings concerning *ATP8B1* and *ABCB11* genes, histopathology, and treatment response were examined retrospectively. Results: Seven Japanese patients with BRIC were enrolled (four male, three female; four BRIC-1 and three BRIC-2). The median age at onset for BRIC-1 was 12 years; for BRIC-2, it was 1 month. Intermittent cholestatic attacks numbered from one to eight during the 11 years of median follow-up. Six patients received a mainstream education; only one patient attended special education. None developed cirrhosis. Three with BRIC-1 showed compound heterozygosity for a variant *ATP8B1* gene, while one was heterozygous; two BRIC-2 patients showed compound heterozygosity in *ABCB11* and one was heterozygous. Liver biopsy specimens obtained during cholestatic attacks showed fibrosis varying from none to moderate; inflammation was absent or mild. Rifampicin administered to three patients for cholestatic attacks was effective in all, as was cholestyramine in two of three. Conclusions: To our knowledge, this is the first East Asian multicenter study of BRIC patients. Onset age and number of cholestatic attacks varied. Rifampicin and cholestyramine were effective against attacks. No patient developed cirrhosis; most had normal growth and development. The long-term outcomes were satisfactory.

## 1. Introduction

Benign recurrent intrahepatic cholestasis (BRIC) is a rare autosomal recessive inherited liver disease associated with canalicular transport defects causing a mild form of progressive familial intrahepatic cholestasis (PFIC). First described by Summerskill and Walshe in 1959, BRIC is characterized by intermittent episodes of cholestasis without progression to liver failure [1,2,3]. While typical PFIC presents in infancy or early childhood and often leads to liver cirrhosis, BRIC usually appears later and has a relatively benign course [3]. BRIC is subdivided into type 1 (BRIC-1) and type 2 (BRIC-2) according to the causative gene. Respectively, BRIC-1 and BRIC-2 are caused by pathogenic variants in the ATPase Phospholipid-Transporting 8B1 (*ATP8B1*) and ATP-Binding Cassette Subfamily B Member 11 (*ABCB11*) genes [4,5]. Symptoms include recurrent jaundice and pruritus, and laboratory blood tests show cholestasis with elevated serum direct bilirubin (D Bil) and total bile acids (TBAs), but often normal or mildly elevated alanine aminotransferase (ALT) and γ-glutamyltransferase (GGT) [3]. Cholestatic symptoms may persist for several weeks to months, but the episodes resolve spontaneously without progression to liver failure. Liver function test results are normal during the asymptomatic period [3]. Treatment is supportive, aiming to relieve pruritus and other symptoms of cholestasis until spontaneous resolution of the cholestatic attack [2,3]. As BRIC is very rare in Japan and other East Asian countries, the paucity of clinical reports from this region has led to uncertainty about clinical, genetic, and prognostic features in East Asian patients. The present multicenter study aims to describe these attributes of BRIC in Japanese patients.

## 2. Methods

### 2.1. Design and Ethical Matters

We conducted a multicenter, retrospective, observational study which complied with the ethical guidelines of the Declaration of Helsinki of 1975, as revised in 2013, and was approved by the Ethical Committee of Kurume University and its counterparts at other participating centers. This retrospective study was conducted using an opt-out method.

### 2.2. Study Subjects

Subjects were retrospectively enrolled patients with a clinical and genetic diagnosis of BRIC-1 or BRIC-2 who visited 4 pediatric centers and 1 adult center in Japan between April 2007 and March 2022. BRIC was defined by one or more intermittent cholestatic attacks (serum D Bil > 1.5 mg/dL) with symptoms such as jaundice or pruritus associated with biallelic variants in the *ATP8B1* or *ACBC11* gene, or alternatively two or more intermittent cholestatic attacks associated with at least a single allelic variant in the *ATP8B1* or *ACBC11* gene. Patients had no symptoms of cholestasis such as jaundice or pruritus except during a cholestatic attack. Demographic attributes, clinical course, laboratory blood tests, molecular genetic results concerning the *ATP8B1* or *ABCB11* gene, histopathologic findings in the liver, and details of treatment were ascertained retrospectively. Laboratory blood tests included serum ALT, GGT, total bilirubin (T Bil), D Bil, and TBA. Growth was evaluated according to height and weight, expressed in terms of standard deviation (SD) [6]. Efficacy of treatment was defined as follows: improvement and cessation of a cholestatic attack; partial improvement and reduction in pruritus; or no improvement and lack of a response or deterioration of symptoms [2].

### 2.3. Molecular Genetic Analysis of ATP8B1 or ABCB11

We examined variants in patients’ *ATP8B1* or *ABCB11* genes detected via either targeted sequencing analysis or targeted next-generation sequencing using a previously reported custom-configured neonatal/infantile intrahepatic cholestasis gene panel including both genes [7,8,9,10]. The pathogenicity of detected variants was evaluated according to the American College of Medical Genetics and Genomics’ interpretation guidelines. Sequences were compared with the reference sequence (GenBank accession number NM_ 005603.6 for *ATP8B1* and NM_003742.4 for *ABCB11*).

### 2.4. Liver Histopathology

Liver biopsy specimens obtained from patients during a cholestatic attack were assessed pathologically based on the New Inuyama Classification of chronic hepatitis. In this classification, chronic hepatic disease is characterized according to the degree of fibrosis (F) as follows: F0 (no fibrosis, equivalent to Ishak stage 0), F1 (fibrosis evident as portal expansion, equivalent to Ishak stage 1 to 2), F2 (bridging fibrosis, equivalent to Ishak stage 3), F3 (bridging fibrosis with lobular distortion, equivalent to Ishak stage 4), or F4 (cirrhosis, equivalent to Ishak stage 5 to 6). Additionally, the New Inuyama Classification assesses chronic hepatic disease activity (A) based on the degree of lymphocytic infiltration and necrosis of hepatocytes: A0 (no necro-inflammatory reaction), A1 (mild necro-inflammatory reaction), A2 (moderate necro-inflammatory reaction), and A3 (severe necro-inflammatory reaction) [11,12]. Immunohistochemical assessments of bile acid transport pump protein (BSEP; Santa Cruz Bio-technology, Santa Cruz, CA, USA) and multi-drug resistance protein 2 (MRP2; Enzo Life Sciences, Farmingdale, NY, USA) were performed for some patients. The expressions of BSEP and MRP2 were compared with normal controls and characterized as normal, strong, weak, or absent.

### 2.5. Statistical Analysis

Continuous variables are expressed as medians with minimum and maximum, while categorical variables are expressed as percentages. The Mann–Whitney U test was used where appropriate. Variables with 2-sided *p* values below 0.05 were considered to indicate statistical significance. Statistical analyses were carried out using GraphPad Prism (version 9.1.0; GraphPad Software), which was also used to produce related figures.

## 3. Results

### 3.1. Patient Characteristics and Long-Term Outcome

We enrolled seven Japanese BRIC patients (four male and three female; four BRIC-1 and three BRIC-2). The characteristics of the patients are shown in Table 1. Pathogenic gene variants and portions of clinical information concerning Patients 2, 3, and 7, as well as pathogenic gene variants for Patients 1, 4, and 5, have been reported previously [7,8,13,14,15]. The median age at onset for patients with BRIC-1 was 12 years (range, 9 months to 21 years), while that for BRIC-2 was 1 month (1 month to 6 months). The number of intermittent cholestatic attacks varied from one to eight during the follow-up period (median, 11 years: 4 to 24). The durations of individual intermittent cholestatic attacks ranged from 2 weeks to 8 months (median, 2 months). Most cholestatic attacks had no known triggering event, while a few were preceded by infection with Epstein–Barr virus, cytomegalovirus, or influenza virus. Four patients had coexisting or concurrent disorders including atopic dermatitis, type 2 diabetes, intracranial hemorrhage, or developmental issues. All patients had normal growth parameters except for Patient 4, an adult BRIC-1 patient with type 2 diabetes whose weight was 3.2 SD above the mean (body mass index, 34.8). Six patients received a regular education; only one patient attended special education. Two of three adult patients were employed. No patient developed cirrhosis during their clinical course; the attending physicians evaluated liver function via physical examination, blood tests including platelet count and prothrombin time, transient elastography, and imaging, including ultrasonography and computed tomography.

### 3.2. Genetic Features

The pathogenic gene variants in each patient are shown in Table 1. Three patients with BRIC-1 had a compound heterozygous variant in the *ATP8B1* gene, while another had only one allele variant identified. Two patients with BRIC-2 had compound heterozygosity involving the *ABCB11* gene, while another had only one allele variant identified. Seven pathogenic variants were identified in the *ATP8B1* gene and five were detected in the *ABCB11* gene. The *ATP8B1* variants included two missenses, one nonsense, three deletions (frameshifts), and one splice-site change; the *ABCB11* variants included three missenses and two nonsenses. The variants all differed among patients, with no high-frequency variants identified.

### 3.3. Symptoms and Liver Function Changes during Cholestatic Attacks

Jaundice and pruritus occurred in 100% of patients during attacks (7/7), while gray stool discoloration, fatigue, anorexia, and weight loss were noted in 57% (4/7), 29% (2/7), 29% (2/7), and 14% (1/7), respectively. No patient had any symptom between attacks. When we compared the serum liver function test results between cholestatic attacks and non-attack evaluations, serum T Bil peaked during attacks (median, 12.0 mg/dL; range, 7.5 to 27.3), as did D Bil (8.3 mg/dL; 5.6 to 21.0) and TBA (308 μmol/L; 238 to 425), while ALT (30 U/L; 22 to 93) and GGT (25 U/L; 8 to 60) ranged from normal to mildly elevated. Between cholestatic attacks, ALT (18 U/L; 13 to 33), GGT (12 U/L; 7 to 13), T Bil (0.7 mg/dL; 0.4 to 0.9), D Bil (0.1 mg/dL; 0.1 to 0.5), and TBA (3 μmol/L; 1 to 7) showed nearly normal values. Serum T Bil, D Bil, and TBA showed significant differences between attack and non-attack periods, while attack-related changes in ALT and GGT fell short of significance (*p* = 0.0682 and *p* = 0.0921, respectively; Figure 1).

### 3.4. Histopathologic Findings in the Liver

Liver biopsy specimens were obtained from all seven BRIC patients during a cholestatic attack. The liver histopathologic findings are shown in Table 2. The fibrosis stage among patients ranged from F0 to F3: F0, 43% (3/7); F1, 29% (2/7); F2, 14% (1/7); and F3, 14% (1/7). The fibrosis stages in patients with BRIC-2 were more advanced (F1 to F3) than in those with BRIC-1 (F0 and F1). Inflammation was absent or mild in all patients: in 57% of patients (4/7) it was graded as A0, while in 43% (3/7) it was graded as A1. An immunohistochemical evaluation of BSEP and MRP2 was carried out in five patients (three BRIC-1 and two BRIC-2). BSEP expression was strong in three patients with BRIC-1 but weak in two patients with BRIC-2. MRP2 expression was strong in all five patients. Figure 2 shows the liver histopathologic findings during a cholestatic episode in a patient with BRIC-1 (Patient 1).

### 3.5. Treatment for Cholestatic Attacks

Treatments for cholestatic attacks are summarized in Table 3. Ursodeoxycholic acid (UDCA), phenobarbital, rifampicin, cholestyramine, antihistamines, prednisolone, 4-phenylbutyrate, and nasobiliary drainage were, respectively, administered to 100% (7/7), 57% (4/7), 43% (3/7), 43% (3/7), 43% (3/7), 14% (1/7), 14% (1/7), and 14% (1/7) of patients as treatment for cholestatic attacks. Several patients received multiple drugs which were used simultaneously. Rifampicin was effective in all three patients to whom it was given, and cholestyramine was effective in two of three patients who received it. UDCA, phenobarbital, and antihistamines showed poor efficacy overall, while prednisolone, 4-phenylbutyrate, and nasobiliary drainage could not be meaningfully evaluated because only one patient was treated with them. Six of the seven patients received no treatment in the absence of cholestasis. The other patient (Patient 6) continued UDCA treatment during the absence of cholestasis because it was effective against his cholestatic attacks until he had an episode after attempting discontinuation of UDCA. No patient underwent plasmapheresis or liver transplantation.

## 4. Discussion

In the present study, we characterized BRIC in seven Japanese patients with respect to clinical and genetic features, treatment choices, and efficacy against cholestatic attacks, as well as long-term outcomes.

Among patients with BRIC, the first cholestatic attack often occurs in childhood or early adolescence, although attacks may begin in infancy or late middle age [3]. In the present study, age at first cholestatic attack ranged from 1 month to 21 years; BRIC-2 was associated with younger clinical onset than BRIC-1. Likewise, Davit-Spraul et al. [16] report that patients with a more severe disorder, PFIC type 2, begin to show signs of cholestasis earlier than those with PFIC type 1. While age at clinical onset in patients with BRIC can range from infancy to adulthood, initial symptoms typically occur earlier in BRIC-2 than BRIC-1.

The duration of a cholestatic attack among patients with BRIC ranges from weeks to months, while the frequency of attacks ranges from several times a year to once per decade. In the present study, the median duration of intermittent cholestatic attacks was 2 months, with a frequency varying from two attacks per year to one per decade. During cholestatic attacks, all patients had jaundice and pruritus, while some also had various symptoms such as grayish stools, fatigue, anorexia, and weight loss. Laboratory blood tests during attacks showed elevated serum T Bil, D Bil, and TBA with normal to mildly elevated ALT and GGT, with normalization between attacks. No patient developed cirrhosis. Most had normal growth and development. Six of the seven patients received a mainstream education, while two of three adults were employed. Overall, the long-term outcomes in Japanese patients with BRIC were satisfactory.

Genetic testing showed that three patients among the four with BRIC-1 had compound heterozygosity for variants in the *ATP8B1* gene, and one was heterozygous. Two patients with BRIC-2 had compound heterozygosity in the *ABCB11* gene while one was simply heterozygous. BRIC, like PFIC, is considered to be an autosomal recessive disorder, but some patients have only a single allelic variant in the *ATP8B1* or *ABCB11* gene [4,5]. In our present study, two of seven Japanese patients with BRIC proved to only have an allelic variant in the *ATP8B1* or *ABCB11* gene. Altogether, we identified 12 pathogenic variants in seven Japanese BRIC patients. The *ATP8B1* variants included two missenses, one nonsense, three deletions (frameshifts), and one splice-site change, while the *ABCB11* variants included three missenses and two nonsenses. Klomp et al. [4] and van Mil et al. [5] reported that missense variants are more common in patients with BRIC while nonsense, frameshift, and large deletion variants are more common in PFIC. Interestingly, 42% (5/12) of variants in the present Japanese study were missense, while the other 58% (7/12) included nonsense, deletion (frameshift), and splice-site changes. Additionally, variants in our Japanese study were all different, with no single variant occurring at a high frequency, in contrast to some previous BRIC studies from Western countries that reported some variants to be relatively common [4,5]. The differing makeups of the genetic results may reflect racial differences between Western and East Asian populations.

Our multicenter BRIC study included liver histopathologic findings in all patients. We found the degree of liver fibrosis to vary among patients from none to moderate, while histologic features of inflammation during cholestatic attacks were absent or mild. While most patients did not undergo a follow-up liver biopsy when cholestatic attacks had resolved, liver function blood test results, transient elastographic findings, and images including ultrasonography and computed tomography were normal at these times. No patients developed cirrhosis. In our results, intermittent cholestatic attacks in patients with BRIC were sometimes associated with mild to moderate fibrosis without progression to cirrhosis. However, careful long-term monitoring of patients with BRIC is needed, considering that some patients with a form of disease intermediate between BRIC and PFIC have experienced a delayed development of cirrhosis [4,17,18].

We also performed immunohistochemical examinations for BSEP and MRP2 in the liver biopsy specimens. BSEP participates in the transport of bile salts from hepatocytes to bile canaliculi, while MRP2 facilitates the elimination of conjugated bilirubin [2]. A strong expression of MRP2 (and BSEP in BRIC-1) suggests an accelerated secretion of conjugated bilirubin (and bile salts in BRIC-1) from hepatocytes to bile canaliculi during attacks.

Drugs and invasive measures have both been used to treat cholestatic episodes in patients with BRIC. Drugs include UDCA, phenobarbital, rifampicin, cholestyramine, antihistamines, prednisolone, and 4-phenylbutyrate, while invasive procedures include biliary drainage and plasmapheresis. Rifampicin has been reported to be the drug most effective against cholestatic attacks in patients with BRIC [2,7]. In our present study, rifampicin was effective in all three patients who received it. Hepatocytic mechanisms underlying rifampicin efficacy include an enhancement in MRP2 expression and activation of the enzymes uridine diphosphate glucuronosyltransferase 1A1 and cytochrome P450 3A4. The latter effect involves the activation of pregnane X receptor-regulated transcription of the cytochrome, which participates in the 6α-hydroxylation of bile acids. After this hydroxylation, bile acids can be excreted through the basolateral membrane via MRP3 and/or MRP4, with subsequent excretion via urine [7,19,20]. Recently, Koukoulioti et al. [21] reported that cholestyramine might not only shorten icteric episodes in BRIC-1, but also prevent their onset. Cholestyramine is an anion exchange resin considered to accelerate the excretion of bile acids by promoting their adsorption in the intestine, and ultimately their excretion in feces. As a result, the enterohepatic circulation of bile acids is reduced and the bile acid pool is decreased [22,23]. Cholestyramine was effective in two of three patients in our study who received it. Most patients required no treatment during the absence of cholestasis. Rifampicin and cholestyramine appear to represent key agents for decreasing or ameliorating cholestatic attacks in patients with BRIC.

Although our present study offers the strength of a multicenter study, a number of limitations are evident. First, relatively small numbers of BRIC subjects were enrolled because BRIC is a very rare disease in Japan. Second, we could not assess liver histopathology, including stage of fibrosis, in the absence of cholestasis because no follow-up liver biopsy was performed in most patients. Third, since our subjects were all Japanese, our findings might not be generalizable to patients in other East Asian countries or different ethnic groups. Prospective multicenter controlled studies including larger and more diverse populations might provide further information.

In conclusion, we retrospectively clarified clinical, pathologic, and genetic features, treatment for cholestatic attacks, and long-term outcomes in Japanese patients with BRIC. To our knowledge, this is the first multicenter study of BRIC patients from East Asia. The age at onset and number of cholestatic attacks varied. Both rifampicin and cholestyramine were effective against cholestatic attacks. No patient developed cirrhosis, and most patients attained normal growth and development. The long-term outcomes among Japanese BRIC patients were satisfactory.

## Figures and Tables

**Figure 1 jcm-12-05979-f001:**
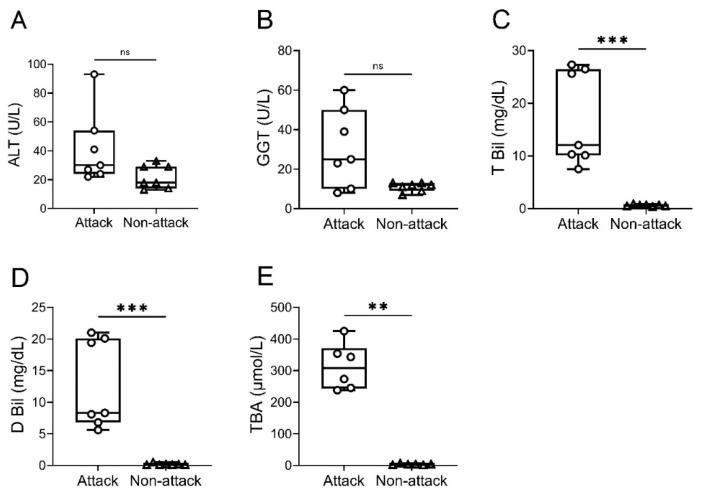
**Liver function blood test results compared between cholestatic attacks and quiescent periods.** Concentrations of serum alanine aminotransferase (ALT; panel (**A**)), γ-glutamyltransferase (GGT; panel (**B**)), total bilirubin (T Bil; panel (**C**)), direct bilirubin (D Bil; panel (**D**)), and total bile acids (TBAs; panel (**E**)) were compared between cholestatic attacks (Attack) and periods without attacks (Non-attack). Horizontal lines in the middle of boxes indicate medians; tops and bottoms of boxes correspond to 75th and 25th percentiles, respectively. Whiskers above and below boxes give maximum and minimum concentrations, respectively. Analyses for each variable include 7 patients, except for TBA (6 patients). ns, not significant; **, *p* < 0.01; ***, *p* < 0.001.

**Figure 2 jcm-12-05979-f002:**
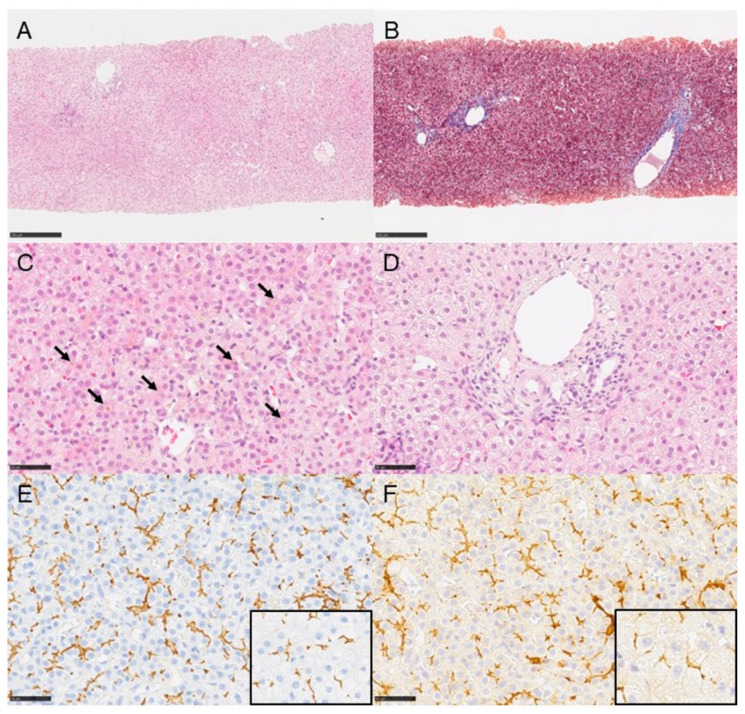
**Histopathologic findings in the liver in Patient 1.** Liver tissues show no significant changes in lobular architecture (panel (**A**); hematoxylin and eosin stain) and no fibrosis (panel (**B**); Masson trichrome stain). Intrahepatic cholestasis is unaccompanied by giant cell transformation, and bile plugs are present in bile canaliculi (panel (**C**); hematoxylin and eosin stain; arrows, bile plugs). No fibrosis or inflammation are present in portal areas (panel (**D**); hematoxylin and eosin stain). Strong immunohistochemical staining is present for bile salt export pump (panel (**E**)) and multi-drug resistance protein 2 (panel (**F**)) along bile canalicular membranes, in contrast with normal controls (insets). Scale bars represent 250 μm in panels (**A**,**B**) and 50 μm in panels (**C**–**F**).

**Table 1 jcm-12-05979-t001:** Patient characteristics, genetic features, and long-term outcomes.

Patient No.	Disease Type	Sex	Age at Onset	Age at Recent Visit (y)	Duration of Follow-Up (y)	Number of Cholestatic Attacks	Pathogenic Variants in *ATP8B1* or *ABCB11* Gene (Allele 1/Allele 2)	Other Complications and Problems (Age at Diagnosis)	Height/Weight at Recent Visit (SD)	School/ Employment	Ref *
							*ATP8B1*				
1	BRIC-1	F	9 m	8	8	2	c.922G>A;p.(Gly308Ser) /c.3579_3589del;p.(Arg1194Valfs*38)		1.5/1.2	Mainstream education	[8,14]
2	BRIC-1	F	7 y	11	4	1	c.1227del;p.(Val410*) /c.2212del;p.(Thr738Leufs*5)		−0.5/−1	Mainstream education	[7]
3	BRIC-1	M	17 y	21	4	8	c.1429+2t>g /not detected	Atopic dermatitis (4 y)	0.4/−0.3	Mainstream education/ employed	[13]
4	BRIC-1	F	21 y	43	22	3	c.1408T>G;p.(Cys470Gly) /c.2854C>T;p.(Arg952*)	Type 2 diabetes (30 y)	−0.2/3.2	Mainstream education/ employed	[14]
							*ABCB11*				
5	BRIC-2	M	1 m	11	11	3	c.3121T>C;p.(Tyr1041His) /c.3904G>T;p.(Glu1302*)		0.6/−0.4	Mainstream education	[8]
6	BRIC-2	M	1 m	18	18	3	c.1723C>T;p.(Arg575*) /c.1907A>G;p.(Glu636Gly)	Intracranial hemorrhage (2 m)	−0.2/−0.1	Mainstream education	
7	BRIC-2	M	6 m	24	24	8	c.1211A>G;p.(Asp404Gly) /not detected	Developmental disorders (17 y)	0.4/−0.5	Special education/ unemployed	[15]

No., number; y, year(s); SD, standard deviation; Ref, reference(s); BRIC, benign recurrent intrahepatic cholestasis; F, female; m, month(s); M, male. Sequences were compared with their reference sequences (GenBank accession number NM_ 005603.6 for ATP8B1 and NM_003742.4 for ABCB11). *, pathogenic variants and some of the clinical information for Patients 2, 3, and 7 have been previously reported, as well as pathogenic variants for Patients 1, 4, and 5.

**Table 2 jcm-12-05979-t002:** Summary of liver histopathologic findings during cholestatic attacks.

Patient No.	Disease Type	Age at Liver Biopsy	Fibrosis Stage	Inflammation Grade	BSEP Expression	MRP2 Expression
1	BRIC-1	9 m	F0	A0	Strong	Strong
2	BRIC-1	7 y	F1	A0	Strong	Strong
3	BRIC-1	17 y	F0	A0	Strong	Strong
4	BRIC-1	39 y	F0	A0	ND	ND
5	BRIC-2	3 y	F1	A1	Weak	Strong
6	BRIC-2	3 y	F3	A1	ND	ND
7	BRIC-2	17 y	F2	A1	Weak	Strong

BSEP, bile acid transport pump protein; MRP2, multi-drug resistance protein 2; BRIC, benign recurrent intrahepatic cholestasis; m, months; y, years; ND, not done; F0, no fibrosis; F1, fibrosis with portal expansion; F2, bridging fibrosis; F3, bridging fibrosis with lobular distortion; A0, no necro-inflammatory reaction; A1, mild necro-inflammatory reaction.

**Table 3 jcm-12-05979-t003:** Treatments for cholestatic attacks.

Drugs	Number of Patients	Outcome (Number of Patients)
Ursodeoxycholic acid	7	Partial improvement (2) No improvement (5)
Phenobarbital	4	Partial improvement (1) No improvement (3)
Rifampicin	3	Improvement (2) Partial improvement (1)
Cholestyramine	3	Partial improvement (2) No improvement (1)
Antihistamine	3	Partial improvement (1) No improvement (2)
Prednisolone	1	Partial improvement (1)
4-phenylbutyrate	1	Partial improvement (1)
Nasobiliary drainage	1	No improvement (1)

Improvement indicates abortion of a cholestatic attack. Partial improvement indicates reduction in pruritus. No improvement indicates no change or worsened symptoms.

## Data Availability

The data leading to the findings of this study are available upon request from the corresponding author.

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
