# Peer review of "Clinicopathologic Features, Genetics, Treatment, and Long-Term Outcomes in Japanese Children and Young Adults with Benign Recurrent Intrahepatic Cholestasis: A Multicenter Study"

_jcm, 2023, doi:10.3390/jcm12185979_

Round 1

Reviewer 1 Report

Thank you this is a good quality report of a case series of patients with BRIC. 

I only have few suggestions:

1. Table 1 and elsewhere: "general education" best to replace with "mainstream education". 

2. line 158 page 4. "simple heterozygous" best to replace here and elsewhere: "only one mutant allele identified". This is because it is possible that the second mutation exists but has not been detected due to difficult sequence or deep intronic mutation. 

3. line 264 page 8 "all told" replace with "altogether". 

4. Were identified mutations reported before and what was the phenotype associated with them?

no other comments. 

Author Response

Journal of Clinical Medicine Editorial Office

Dear Editors and Reviewers,

RE: jcm-2558703

Title: Clinicopathologic features, genetics, treatment, and long-term outcome in Japanese children and young adults with benign recurrent intrahepatic cholestasis: a multicenter study

Thank you very much for your encouraging letter of September 5, 2023. Based on your kind suggestions and those of reviewers concerning our manuscript numbered jcm-2558703 and titled "Clinicopathologic features, genetics, treatment, and long-term outcome in Japanese children and young adults with benign recurrent intrahepatic cholestasis: a multicenter study," my coauthors and I have made revisions as summarized below. We also entered all reviewer-prompted changes in the revised manuscript in red with yellow highlighted. When we could not carry out a suggestion, we explained the reasons for difficulty in this letter.

Response to Reviewer #1’s comments:

  1. Table 1 and elsewhere: "general education" best to replace with "mainstream education".

We changed “general education” to “mainstream education.”

  1. line 158 page 4. "simple heterozygous" best to replace here and elsewhere: "only one mutant allele identified". This is because it is possible that the second mutation exists but has not been detected due to difficult sequence or deep intronic mutation.

We changed "simple heterozygous" to "only one variant allele identified."

  1. line 264 page 8 "all told" replace with "altogether".

We changed "all told" to "altogether."

  1. Were identified mutations reported before and what was the phenotype associated with them?

All identified variants in this manuscript were previously reported. We cited previous reports including those variants in Table 1 and Reference. Most previous reports described those variants did not clarify long-term clinical course and it is difficult to accurately diagnose the phenotype of whether it is PFIC or BRIC. In our manuscript, we clarify long-term outcome in Japanese BRIC patients and believe that the genotypes in this paper may help to predict genotype-phenotype correlation in the present and future BRIC patients.

We attached the revised manuscript.

Reviewer 2 Report

This study provides a well-organized description and analysis of patients with benign recurrent intrahepatic cholestasis in Japan. However, there are also some concerns that the authors need to address.

Page 7, line 215 : Treatment for cholestatic attacks

Treatment of patients with cholestasis may vary depending on the physician's judgment. Still, there may be cases where drugs such as UDCA, phenobarbital, rifampicin, and cholestyramine, as mentioned by the author, are prescribed simultaneously. Wouldn't it be challenging to accurately assess the efficacy of a drug when multiple drugs are used simultaneously?

I believe that most other sentences and compositions in the article are suitable and can benefit other readers.

Author Response

Journal of Clinical Medicine Editorial Office

Dear Editors and Reviewers,

RE: jcm-2558703

Title: Clinicopathologic features, genetics, treatment, and long-term outcome in Japanese children and young adults with benign recurrent intrahepatic cholestasis: a multicenter study

Thank you very much for your encouraging letter of September 5, 2023. Based on your kind suggestions and those of reviewers concerning our manuscript numbered jcm-2558703 and titled "Clinicopathologic features, genetics, treatment, and long-term outcome in Japanese children and young adults with benign recurrent intrahepatic cholestasis: a multicenter study," my coauthors and I have made revisions as summarized below. We also entered all reviewer-prompted changes in the revised manuscript in red with yellow highlighted. When we could not carry out a suggestion, we explained the reasons for difficulty in this letter.

Response to Reviewer #2’s comments:

Page 7, line 215: Treatment for cholestatic attacks

Treatment of patients with cholestasis may vary depending on the physician's judgment. Still, there may be cases where drugs such as UDCA, phenobarbital, rifampicin, and cholestyramine, as mentioned by the author, are prescribed simultaneously. Wouldn't it be challenging to accurately assess the efficacy of a drug when multiple drugs are used simultaneously?

Thank you very much for your insightful suggestion. We agree with you and think it is one of the study limitations. We added the sentence of "Several patients received multiple drugs which were used simultaneously" in the paragraph of Treatment for cholestatic attacks.

We attached the revised manuscript.
